# The Interaction Network and Signaling Specificity of Two-Component System in *Arabidopsis*

**DOI:** 10.3390/ijms21144898

**Published:** 2020-07-11

**Authors:** Ruxue Huo, Zhenning Liu, Xiaolin Yu, Zongyun Li

**Affiliations:** 1Institute of Integrative Plant Biology, Jiangsu Key Laboratory of Phylogenomics & Comparative Genomics, School of Life Sciences, Jiangsu Normal University, Xuzhou 221116, China; huoruxue@lyu.edu.cn; 2College of Agriculture and Forestry Sciences, Linyi University, Linyi 276000, China; 3College of Agriculture and Biotechnology, Zhejiang University, Hangzhou 310058, China; xlyu@zju.edu.cn

**Keywords:** two-component system, multi-step phosphorelay, histidine kinases, histidine-containing phosphotransfer proteins, response regulators, protein–protein interaction network, signaling specificity

## Abstract

Two-component systems (TCS) in plants have evolved into a more complicated multi-step phosphorelay (MSP) pathway, which employs histidine kinases (HKs), histidine-containing phosphotransfer proteins (HPts), and response regulators (RRs) to regulate various aspects of plant growth and development. How plants perceive the external signals, then integrate and transduce the secondary signals specifically to the desired destination, is a fundamental characteristic of the MSP signaling network. The TCS elements involved in the MSP pathway and molecular mechanisms of signal transduction have been best understood in the model plant *Arabidopsis thaliana*. In this review, we focus on updated knowledge on TCS signal transduction in *Arabidopsis*. We first present a brief description of the TCS elements; then, the protein–protein interaction network is established. Finally, we discuss the possible molecular mechanisms involved in the specificity of the MSP signaling at the mRNA and protein levels.

## 1. Introduction

In all living organisms, responses to specific environmental stimuli are mediated by complicated signal transduction pathways. Protein phosphorylation is a major mechanism regulating signal transduction pathways in both the prokaryotes and eukaryotes. Using ATP as the phosphate donor, protein kinases can catalyze phosphorylation of either themselves or the other protein substrates at specific serine, threonine, tyrosine, or histidine residues. Based on substrate specificity, protein kinases could be classified into 3 main categories: Serine–threonine (ser–thr) kinases, tyrosine (tyr) kinases, and histidine kinases (HKs) [1]. In eukaryotes, ser–thr kinases and tyr kinases are the predominant phosphorylation approaches. However, in bacteria, phosphorylation relies upon histidine kinases [2]. In response to an environmental stimulus, the histidine kinase perceives the signal and autophosphorylates its conserved histidine (His) residue, which is then transferred to a conserved aspartic acid (Asp) residue within another group of signal transducers called the response regulators (RRs). RRs are transcription factors, and phosphorylation confers upon the ability to mediate downstream signaling, and thus converting an external stimulus into an internal reference signal. This type of signal transduction is designated as the two-component system (TCS) [3,4].

Two-component signaling elements were originally identified in *Escherichia coli* (*E. coli*) when studying the nitrogen regulatory protein system [5]. The one-step TCS described previously is a major signal transduction pathway in prokaryotes. Two-component signaling elements have also been identified in yeasts, fungi, slime molds, and plants [6,7,8,9,10,11]. In eukaryotes, the basic two-component scheme has elaborated to become a more complicated multi-step phosphorylation system, which employs a hybrid HK with both HK and Rec (receiver) domains, a histidine-containing phosphotransfer (HPt) protein, and a separate RR (Figure 1 and Figure 2) [4]. The HPt protein is responsible for phosphate transfer between the hybrid HK and RR. In this multi-step phosphorelay, the sequence of phosphate transfer is His–Asp–His–Asp, which could provide more choices for signal transduction and contribute to increasing the diversity and complexity of the regulation [12]. Beside the canonical two-component circuitry, cytokinin response factors (CRFs), which act downstream of HPt proteins to mediate cytokinin-regulated gene expression, define a new branch point in the two-component system signal transduction pathway [13].

To date, elements of the TCS, including hybrid HKs, HPts, and RRs, have been identified at a genome-wide scale in *Arabidopsis* [3], rice [14,15], tomato [16], soybean [17], maize [18,19], *Physcomitrella patens* [20], and *Brassica rapa* [21], among others. However, the exact roles and signal transduction mechanisms of the TCS elements in plants are largely undetermined, and *Arabidopsis* represents the best understood TCS element in plants. This review will focus on updated knowledge on TCS signal transduction in *Arabidopsis*. We first provide a brief description of the component elements, then establish a protein–protein interaction network, and discuss possible molecular mechanisms of signaling specificity in multi-step phosphorelay pathways.

## 2. Two-Component Signaling Elements in *Arabidopsis*

Genome-wide analysis has revealed that there are a total of 67 two-component signaling elements in *Arabidopsis*, including 16 HKs (AHKs), 6 HPt proteins (AHPs), 10 Type-A response regulators (Type-A ARRs), 12 Type-B response regulators (Type-B ARRs), 2 Type-C response regulators (Type-C ARRs), 9 pseudo response regulators (pseudo-ARRs), and 12 CRFs.

### 2.1. Histidine Kinases

The *Arabidopsis* HKs are grouped into 3 main families: AHK, ethylene receptor, and phytochrome. The cytokinin receptors are located in a subdivided branch of the AHK family (Figure 3).

#### 2.1.1. AHK Family

The AHK family contains 6 hybrid HKs, AHK1, AHK2, AHK3, AHK4 (CRE1/WOL), AHK5, and CKI1, all with HK domain, HATPase domain, and Rec domain. AHK2, AHK3, and AHK4 are cytokinin receptors with an additional transmembrane (TM) domain and cyclases/histidine kinases associated sensing extracellular (CHASE) domain that functions in cytokinin binding [22,23], which mainly show subcellular localization in the endoplasmic reticulum (ER) and plasma membrane (PM) [24,25,26]. The isolation and characterization of T-DNA insertions has demonstrated their roles in diverse cytokinin-regulated developmental processes, including cell division, vascular differentiation, leaf senescence, male and female gametogenesis, seed development, and adverse environmental stress responses [23,27,28,29,30,31,32,33,34]. Most of the single and double T-DNA insertion mutants showed reduced sensitivity to exogenous cytokinin, but no obvious developmental defects were observed under normal growth conditions. However, in *ahk2 ahk3 cre1* triple mutants, response to exogenous cytokinin was almost completely blocked, the growth of roots and leaves was retarded, and fertility was severely impaired in the reproductive growth phase [23,27,30,32], indicating that the 3 cytokinin receptors act as positive regulators in cytokinin signal transduction pathways and show high redundancy in regulatory functions.

AHK1 contains 2 TM domains in the N-terminal region and is mainly located in the PM [24]. Similar to the bacterial and yeast osmosensing pathways, AHK1 was initially proposed to function as a plant osmosensor because of its ability to complement a yeast *sln1-sho1* double mutant lacking its 2 osmosensors and allowed the mutant to grow in high-osmolarity media [35]. However, subsequent studies found that even cytokinin receptors could also complement yeast *SLN1* mutations, suggesting that this function is not unique to AHK1 [31]. Further functional research on AHK1 revealed its roles in stress responses, including drought stress, high salinity stress, and osmotic stress. AHK1 could modulate seed maturation and function downstream of hydrogen peroxide in ABA-induced stomatal closure by regulating the calcium channel activity and calcium oscillation in the guard cells of *Arabidopsis* [31,36,37,38].

CKI1 (cytokinin-independent 1) also contains 2 TM domains in the N-terminal region and is mainly located in the ER and PM [39,40,41]. CKI1 was initially identified in an activation-tagging screen whose ectopic expression conferred cell division and shoot formation on transgenic callus in the absence of exogenous cytokinin [42]. Although overexpression of *CKI1* could confer a typical cytokinin response and induce the expression of cytokinin primary response genes [41], CKI1 fails to bind to cytokinin because of lacking the CHASE domain. Thus, CKI1 is not a cytokinin receptor but it was also found to act upstream of AHP proteins to activate the two-component signaling pathway, in parallel with cytokinin receptors [43,44,45]. *CKI1* loss-of function mutants are female gametophytic lethal and homozygous mutants could not be obtained, suggesting that *CKI1* is required for megagametogenesis [39,44,46,47]. A more detailed study was recently conducted by our team. We introduced female gametophytic cell type-specific single and double markers into a heterozygous *cki1* mutant and found that central cell fate and antipodal cell fate were completely lost and instead transformed into egg cell fate or synergid cell fate. Meanwhile, ectopic expression of *CKI1* in the micropylar end could transform the egg cell fate and synergid cell fate into a central cell fate, indicating that CKI1 is a master regulator of central cell specification [40,48]. Further research on a putative ortholog of *CKI1* in the gymnosperm *Ginkgo biloba* showed that *GbCKI1* could partially rescue the *Arabidopsis cki1* mutant phenotype and activate the TCS signaling pathway, indicating the conserved evolution of *CKI1* function [49]. *CKI1* is also involved in the vegetative growth of *Arabidopsis*. Overexpression of *CKI1* could partially rescue the *ahk2 ahk3* phenotype in vascular tissues, whereas mutation of the CKI1 histidine residue further accentuated mutant phenotypes, indicating CKI1 is also an important regulator of vascular patterning [50].

AHK5, also known as CKI2, lacks the TM domain and is the only HK localized both in the cytoplasm and at the plasmalemma [51]. AHK5 was isolated by the same screen as CKI1, as its overexpression could also induce cytokinin responses in the absence of exogenous cytokinin, and this ability was inheritable [42]. Although no obvious phenotypes were observed in the AHK5 loss-of-function mutant, *ahk5-1*, but root elongation was more sensitive to growth inhibition in response to ethylene; further studies revealed that AHK5 acts as a negative regulator to mediate root elongation in an ETR1-dependent ethylene and ABA signaling pathway [52]. In addition, AHK5 was found to be involved in stomatal responses to endogenous and exogenous signals [51,53], and abiotic and biotic stress responses [54].

#### 2.1.2. Ethylene Receptor Family

Gaseous ethylene is an important hormone involved in plant growth, development, and senescence. Ethylene receptor proteins play essential roles in various developmental processes, including seed germination, leaf senescence, root and shoot elongation, fruit softening, and ripening. The ethylene receptor family of *Arabidopsis* is composed of 5 HK members (ETR1, ETR2, EIN4, ERS1, and ERS2) with ER localization, which also show characteristic features of ethylene-binding TM domains at the N-terminal end and GAF (named after its existence in cGMP-regulated phosphodiesterases and adenylate cyclases of *Anabaena* and the bacterial transcription factor FhlA) protein–protein interaction domain [3,55,56,57,58,59]. These 5 members could be further divided into 2 subfamilies based on the similarities in amino acid sequences and conserved domains [60]. Subfamily I consists of ETR1 and ERS1, both of which show HK activity. Subfamily II includes 3 members (ETR1, ETR2, and EIN4), which contain divergent histidine-kinase-like (HKL) domains lacking conserved histidine residues; thus, they are not involved in HK phosphorylation. Based on in vitro phosphorylation assays, ETR1, ETR2, and EIN4 show ser–thr kinase activity rather than HK activity, suggesting that their phosphorylation is mediated via ser–thr kinases [61,62]. In addition, ETR1, ETR2, and EIN4 contain the Rec domain, whereas this domain is absent in ERS1 and ERS2. ETR1 is a typical hybrid HK, whereas ERS1 is the only known non-hybrid type [63]. The HK activity of the ethylene receptor may not alter the ethylene signal transduction, but evidence showed that the HK activity of ETR1 is important for growth recovery after ethylene removal [64] and endogenous ethylene-regulated promotion of *Arabidopsis* growth [65], suggesting that HK activity could mediate ethylene signaling. Moreover, ethylene signaling and TCS signaling could regulate plant development in a cooperated way. Recent studies showed that ETR1-mediated ethylene signaling could control the root apical meristem through activation of the TCS pathway [66].

#### 2.1.3. Phytochrome Family

Phytochromes are photoreceptors that enable plants to respond to light signals. There are 5 photoreceptor phytochromes, PHYA, PHYB, PHYC, PHYD, and PHYE. Phytochromes are soluble photoreceptor proteins with a PHY domain (light-sensing and chromophore-binding domain) at the N-terminal end, and they can transfer from the cytosol to the cell nucleus when induced by light [67,68]. The middle region is characterized by two PAS (Per/Arndt/Sim) folds. In addition, a divergent HKL domain and HATPase domain are present in the carboxy terminus, but the Rec domain is absent, indicating that these 5 photoreceptors cannot be involved in HK phosphorylation. Studies of *phyb* mutants revealed that the HKL domain is essential for PHYB protein signal transduction; however, removal of this domain does not eliminate PHYB protein activity [69]. Alternatively, plant phytochromes respond to light via ser–thr kinase activity rather than HK activity [70]. Oat (*Avena sativa*) phytochrome can be autophosphorylated on ser/thr residues in a light-dependent manner [71]. Conclusively, plant phytochromes are thought to originate from the ancestral His protein kinase, but have not been reported to be involved in new ser/thr kinase activity [3].

### 2.2. Histidine Phosphotransfer Proteins

The *Arabidopsis* genome encodes 5 HPt domain-containing proteins, designated as AHP1–AHP5 (Figure 4), which contain the highly conserved XHQXKGSSXS motif with a His phosphorylation site. Primary research has shown that AHPs instantly translocate from the cytosol and accumulate in the nucleus upon induction by cytokinin, even though transcripts levels of AHPs were not altered [41]. However, subsequent analysis demonstrated that the subcellular localization of AHPs was independent of cytokinin. Instead, AHPs might maintain a constant cytosolic/nuclear co-localization in the absence or presence of cytokinin [72,73]. AHPs have been shown to act as bridges for multi-step phosphorelays between AHKs and ARRs. In addition, AHPs were found to be able to interact with various AHKs and ARRs based on yeast-two hybrid (Y2H) screens and bimolecular fluorescence complementation (BiFC) assays [53,74,75,76,77,78], consistent with AHPs functions as a mediator. The sensitivity to exogenous cytokinin was not obviously affected for each *ahp* single mutant, but sensitivity was significantly reduced in the *ahp1 ahp2 ahp3* triple mutant. In particular, *ahp1, ahp2, ahp3, ahp4,* and *ahp5* did not respond to cytokinin and were accompanied by severe developmental defects [9,44,79], indicating that AHPs act redundantly as positive regulators in the two-component signaling pathway. AHP6/APHP1 is a pseudo-HPt protein in which the His residue in the phosphorylation site is substituted by an Asn residue; thus, it was not capable of participating in the phosphorelay. Further studies revealed that AHP6 might impair phosphotransfer from AHKs to AHPs by directly competing with functional AHP1–AHP5 to interact with AHKs. The *ahp6* mutant could partially restrain *wol* phenotypes, and the activity of AHP6 is repressed by cytokinin [80,81], suggesting its role as a negative regulator of cytokinin response.

### 2.3. Response Regulators

The *Arabidopsis* genome sequencing project has revealed 24 functional RRs with a Rec domain. Based on phylogenesis, structures, and cytokinin induction properties, these ARRs can be divided into 3 groups: Type-A, Type-B, and Type-C [4]. Besides, there are also 9 genes encoding RRs, which share sequence similarity with typical ARRs in the receiver domain, but they lack the conserved Asp residue and thus were named as pseudo response regulators (PRRs) [82] (Figure 5).

#### 2.3.1. Type-A Response Regulators

The Type-A ARR group is composed of 10 members, ARR3, 4, 5, 6, 7, 8, 9, 15, 16, and 17. Type-A ARRs are relatively small proteins with approximately 230 amino acids, containing a Rec domain along with short N- and C-terminal extensions [83,84,85]. Generally, Type-A ARRs localize to the nucleus, whereas 2 relatively divergent proteins, ARR16 and ARR17, show subcellular localization in the cytoplasm [41,86,87]. Expression of Type-A ARR genes was rapidly upregulated upon induction with exogenous cytokinin and was regarded as cytokinin primary responsive genes [83,84,85,88]. In in vitro assays, Type-A ARRs were confirmed to be involved in the two-component system signaling transduction pathway because of their ability to accept phosphate groups from AHPs. Genetic evidence indicates that Type-A ARR genes function as negative regulators of cytokinin signaling pathways, participating in a negative feedback loop to reduce plant sensitivity to cytokinin [89,90,91,92,93]. Single Type-A ARR gene mutants showed no obvious phenotypes under normal growth conditions, and no obvious changes to exogenous cytokinin response. However, the fact that multiple mutants showed more sensitivity to exogenous cytokinin suggests the functional redundancy of Type-A ARR genes [90].

#### 2.3.2. Type-B Response Regulators

The Type-B ARR group contains 11 members, ARR1, 2, 10, 11, 12, 13, 14, 18, 19, 20, and 21. The Type-B ARRs differ from the Type-A ARRs in that the Type-B ARRs include a Rec domain in the N-terminal end and a long C-terminal extension carrying a Myb-like DNA-binding domain, referred to as the GARP domain and a C-terminal variable region [94,95]. The GARP domain shares common characteristics with MYB transcription factors, indicating that Type-B ARRs could function as transcription factors. The C-terminal variable region is rich in glutamine and proline, a feature frequently observed in transcriptional activator proteins [96]. In addition, the C-terminal variable region was predicted to contain nuclear-localization signals, and Type-B ARRs were examined to show nuclear localization when fused to a reporter protein [41,97]. As transcription factors, Type-B ARRs were found to possess similar DNA-binding motifs, recognizing and binding to a core DNA sequence 5′-(G/A)GAT(T/C)-3′ [98,99]. ARR1 was further confirmed to be able to bind to the promoter region of Type-A ARR genes and activate their transcription both in vitro and in vivo [94,100]. Unlike Type-A ARR genes, mRNA transcripts of Type-B ARR genes were stable when induced with cytokinin [101,102]. However, in transient expression analysis, overexpression of ARR1, ARR2, and ARR10 could enhance ARR6-LUC reporter gene expression, suggesting that Type-B ARRs function as positive regulators in mediating some Type-A ARR gene expression [41].

These 11 Type-B ARR transcription factors could fall into 3 subfamilies according to protein structure and phylogenetic analysis: Type B1 (ARR1, 2, 10, 11, 12, 14, and 18), Type B2 (ARR13 and ARR21), and Type B3 (ARR19 and ARR20) [97]. In addition, ARR23 is a solosist that encodes only 145 amino acids. Although ARR23 contains a typical Rec domain, it might be a pseudo-gene due to the absence of a complete GARP domain [3]. Loss of gene function mutants in Type B1 ARR exhibited reduced sensitivity to cytokinin [13,103], and multiple mutant combinations affected the same cytokinin response as described in *ahk* and *ahp* mutants, including reduced shoot development, aborted primary root growth, malformed hypocotyl elongation, retarded chloroplast development, enlarged seed size, and female gametophyte developmental defects [104,105,106,107,108,109,110], indicating a central role of Type B1 members in cytokinin signaling pathway. In addition, cytokinin-activated Type B1 RRs also play essential roles in hormone crosstalk, as their primary targets are enriched for hormone-related genes [111]. The expression of Type B2 and Type B3 subfamily members are not as broadly prevalent as Type B1 members, and their functional research is mainly restricted to over-expressed lines. Overexpression of ARR21 (Type B2) activated cell proliferation to form callus-like structures in seedlings, and ARR20 (Type B3) over-expressed lines exhibited small flowers and abnormal siliques with reduced fertility [112,113].

#### 2.3.3. Type-C Response Regulators

The Type-C ARR group is composed of 2 members, ARR22 and ARR24. Similar to Type-A ARRs, they contain the Rec domain but without long C-terminal extensions; however, ARR22 and ARR24 share a very low homology with Type-A ARRs based on phylogenetic analysis, and the transcripts of Type-C ARR genes are not regulated by cytokinins [113,114,115]. *ARR22* and *ARR24* are mainly expressed in flowers and siliques, but their corresponding single and double loss-of-function gene mutants do not exhibit any obvious phenotypes compared with the wild-type plants [114,116,117]. However, *ARR22* overexpression lines showed reduced shoot and root growth, reduced cytokinin-responsive gene induction, and increased insensitivity to cytokinin in callus production [114,116,118,119], indicating the possible antagonistic effect of Type-C ARRs on the cytokinin signaling pathway.

#### 2.3.4. Cytokinin Response Factors

Cytokinin response factors are a subfamily containing 12 members (CRF1-CRF12), belonging to a subclade of the AP2/ERF super family (Figure 6), and was named so because of the upregulated gene expression upon cytokinin treatment [13,120,121]. CRFs contain a classical AP2 domain and CRF domain, which act as transcription factors. In addition to transcription regulation, the subcellular localization of CRFs was also regulated by cytokinin. In the protoplast assay, GFP–CRF fusion protein mainly localizes in the cytoplasm and nucleus, but rapidly transfers to the nucleus upon cytokinin treatment [13]. Further analysis showed that cytokinin-induced protein accumulation in the nucleus was dependent on AHKs and AHPs, but was independent of ARRs, suggesting that CRFs may act as a parallel branch of the AHK-AHP-ARR canonical TCS signaling system. The promoters of CRFs contain various binding sites for Type-B ARRs, and the cytokinin induction of CRFs was severely impaired in *arr10 arr12* double mutant, indicating that CRFs may be the direct targets of Type-B ARRs. As CRFs and Type-B ARRs both act downstream of AHKs and AHPs, they may share some common target genes while also having their own specific target genes. Functional studies have shown that *CRFs* are a cluster of plant vascular abundantly expressed genes that are involved in the development of cotyledons, leaves, and embryos, and the regulation process in correlation with various abiotic stresses and phytohormone interaction networks [13,121,122,123,124].

## 3. The Protein–Protein Interaction Network of Two Component System in *Arabidopsis*

Since the discovery of the two-component multi-step phosphorylation system in plants, interaction relationships of the two-component elements were investigated to reveal the mechanisms of the phosphorelays, especially in *Arabidopsis* and hetero- and homodimers were largely screened based on the Y2H, BiFC, and phosphorelay assays. In addition, the protein interaction network was also mapped in rice and polar [125,126]. Here, we focus on the recent progress in the TCS protein interaction relationships, and an interaction network map was further depicted for the TCS elements in *Arabidopsis* (Figure 7).

### 3.1. AHK–AHP Protein Interactions

Suzuki et al. developed a highly sensitive *E. coli* two-hybrid assay system to investigate the possible AHK4–AHPs phosphorelay interactions. The results showed for the first time that AHK4 possessed the ability to interact with AHP1, 2, 3, and 5 through an intimate phosphorelay reaction [127]. Dortay et al. conducted an interaction screen of *Arabidopsis* TCS proteins with Y2H, and screened the interactions between the 3 cytokinin receptors and AHP1, 2, 3, and 5, and further confirmed their interactions by in vitro coaffinity purification method [75]. A more recent study showed that the *Arabidopsis* cytokinin receptors AHK2, AHK3, and AHK4 could interact with any of the tested HPts (AHP1-3) without visible preference in both Y2H and BiFC assays [26]. AHK5 interacted with AHPs 1, 2, 3, and 5 but not with AHP4 in Y2H, whereas BiFC signals were only detected among AHK5 and AHPs 1, 2, 5. The interactions between AHK5 and AHP3 are probably yeast artifacts [53]. However, Bauer et al. detected the interactions between AHK5 and all 6 AHPs. Studies showed that AHK5 receiver domain (AHK5_RD_) alone could interact with AHPs 1, 2, 3, 5, and 6 [78]. Whether the discrepancy in the interaction between AHK5 and AHP3 is correlated with the length of AHK5 protein remains to be determined. When the full-length CKI1 protein was co-expressed with HPts (AHP1-6), strong BiFC signals were observed with AHP2, AHP3, and AHP5, whereas a very weak signal was recorded for AHP1; no interaction was observed between AHP4 and AHP6. Further studies showed that CKI1 interacts preferentially with AHP2 and AHP3, but only weakly with AHP5, and CKI1 receiver domain (CKI1_RD_) alone is required and sufficient for specific interactions between the CKI1 and HPt proteins in vivo [77]. ETR1 is the only ethylene-perceptive hybrid HK in *Arabidopsis*, obtained with Y2H screening. Urao et al. initially identified the interactions between CKI1 and AHPs 1, 2, 3 [74]. Further studies showed that ETR1_RD_ interacts with AHPs 1, 2, 3, and 5 [66], further replenishing its interaction partners AHP5. For AHK1, only interaction with AHP2 was detected by the Y2H method at present [74], and possible interactions with other AHPs still need to be studied.

### 3.2. AHP–ARR Protein Interactions

AHP–ARR protein interactions are key steps for phosphorylation signals to transmit from the cytoplasm to the nucleus. The interactions of AHPs with Type-A ARRs, Type-B ARRs, and Type-C ARRs have also been intensively studied. To gain knowledge on the interactions of TCS proteins, Dortay et al. conducted large-scale interaction screens using the Y2H system, and summarized the previously identified and novel protein interactions [75,76]. In recent years, new interactions between AHPs and ARRs have been identified. Two Type-A ARR members, ARR4 and ARR7, were found to have the ability to interact with AHPs 1, 2, 3, and 5, both in vitro and in vivo [53,128]. A type-B ARR member, ARR18, was also found to interact with AHPs 1, 2, 3, and 5 both in vitro and in vivo [129]. ARR22 is a Type-C ARR that specifically interacts with AHP2, AHP3, and AHP5 [117]. Notably, not all ARR proteins were separately surveyed to identify their AHP interaction partners, even though several members, including ARR12, 13, 17, 19, 20, 21, and 24 were not studied; thus, more evidence is necessary to reveal their interaction relationships and their possible roles in TCS signaling pathways.

### 3.3. AHP-CRF–ARR Protein Interactions

CRFs act downstream of AHPs to target common Type-B ARRs of TCS and some unique pathways. Interaction analysis showed that CRF1-CRF8 could directly interact with AHP1–AHP5 both in vitro and in vivo, except that only CRF2 and CRF3 showed no interaction with AHP2, further demonstrating that the CRFs actually act as a parallel branch of the canonical TCS signaling system [120]. Further analysis of interaction between CRF and ARR showed the interaction of both CRF1 and CRF2 with ARR7 and ARR12, and CRF6-ARR10 interaction was also observed, suggesting a unique ability of CRFs to be involved in specific interactions. The importance of these interactions will have to be examined further to determine their possible roles in the two-component signaling pathway.

## 4. Molecular Mechanisms of Two Component System Signaling Specificity in *Arabidopsis*

Compared with the one-step TCS in prokaryotes, plants have evolved a more complicated MSP pathway, which employs a hybrid HK with an HK and Rec domain, an HPt protein, and a separate RR. The MSP pathways could mediate a wide spectrum of stimulators, including cytokinin, ethylene, and various abiotic stresses. Upon activation of the MSP pathway, histidine-aspartate phosphorelay coupled with multiple protein–protein interactions (HK dimerization, HK–HP interaction, HP dimerization, and HP–RR interaction) to regulate a subset of downstream effectors. The mechanism by which plants perceive the external signals, then integrate and transduce the secondary signals specifically to the desired destination is the fundamental characteristic of the MSP signaling network. To achieve this, plants have evolved several tools. In this review, we focus on mainly 3 different mechanisms for plants to fine-tune TCS signaling: gene expression, protein interaction, and protein modification.

### 4.1. Expression Patterns of Two Component System Related Genes

Gene expression in various tissues and developmental stages are prerequisites for functional proteins. Regulation of gene expression could directly influence cellular protein distribution and function as the first countermeasure mechanism to ensure signaling specificity in the plant MSP network. A more detailed study on the expression patterns of *Arabidopsis* TCS related genes was conducted using large-scale transcriptome sequencing, microarray, semi-quantitative or quantitative reverse transcription polymerase chain reaction (qRT-PCR), and promoter activity assays. A heat map of TCS genes was created based on *Arabidopsis* gene expression data from the AtGenExpress Consortium (*Arabidopsis* eFP Browser) (Figure 8). AHKs together with downstream partners AHPs and ARRs were found in each gene cluster with similar expression patterns, ensuring the fundamental elements essential for phosphorelay signaling processes exist in a certain developmental stage. In addition, a very high degree of overlapping expression occurred at all levels (AHKs, AHPs, and ARRs) of this signaling process, which could explain the high functional redundancy of TCS genes. However, some gene expression discrepancies at the tissue or cellular level may be observed in transgenic lines carrying reporter genes fused with native promoters. Three *Arabidopsis* cytokinin receptors are almost constitutively expressed in the whole plants with preferential expression in the root organs. For a more precise expression dissection, *AHK4* is predominantly expressed in the central cylinder of the root meristem, while *AHK2* and *AHK3* are primarily expressed in the root tip and only slightly overlap with the AHK4 expression domain. In functional complementation analysis, *AHK2* and *AHK3* partially compensated the *ahk4* null phenotypes, but the *AHK2* and *AHK3* activities in the central cylinder were not sufficient to rescue the negative effects on procambial differentiation [130]. Except for the expression profiles of TCS genes, there are signs that the subcellular localization of AHKs is correlated with signaling specificity, and the possibility of multiple sites of perception at PM or ER may determine specific outputs of cytokinin signaling [26].

### 4.2. Protein Dimerizations and Binding Affinity of Two Component System Elements

To ensure TCS protein interaction specificity, a one-to-one interaction relationship between cognate binding partners is indispensable. In bacteria, HK dimerization is normally required for proper functioning and can be used as a tool to finetune TCS signaling. Analogous to bacterial systems, studies have shown that dimerization can occur among HKs, HPs, and RRs in plants. Dimerization refers to a heterodimer where two different proteins interact with each other and a homodimer that self-interact with each other. Here, in the above paragraph, we have described the recent progress on the AHKs-AHPs heterodimers and AHPs-ARRs heterodimers, which are indispensable for phosphorelay from AHKs to AHPs and further AHPs to ARRs. In addition, heterodimers formed within AHKs have been reported. For the cytokinin receptors, studies showed the cytoplasmic region of AHK3 (which includes both the His-containing, phospho-accepting region, and the ATP-binding catalytic domain) could interact with AHK2 and AHK4 [75]. Extensive studies regarding the ability of the ethylene receptors to dimerize have also been performed in *Arabidopsis*, and results showed ETR1 could form heterodimers with all the other ethylene receptors (ERS1, ETR2, ERS2, and EIN4) in pulldown experiments [131]. No AHPs were reported to form heterodimers among AHPs, but further dimerization studies established that hetero-dimerization seems to take place within Type-B ARRs. ARR14 was found to interact with ARR2 [75], but as this interaction was detected with Y2H only, more trials are necessary to explore the possibility of ARRs to form heterodimers. As a parallel branch of TCS, there are solid evidences that *Arabidopsis* CRF1-8 could readily form heterodimers, and even the CRF domain alone is sufficient for CRF–CRF protein interactions [120].

#### 4.2.1. Homo-Dimerization

Except for heterodimers, homodimers widely exist among AHKs, AHPs, and ARRs. The homo-dimerization of AHKs has been reported frequently, including the 3 cytokinin receptors, AHK2, AHK3, and AHK4 [26,75]. Hejatko et al. also found that CKI1 could form homodimers both in vitro and in plants [50]. AHP1, AHP2, and AHP3 were found to form homodimers when co-expressed in *Nicotiana benthamiana* leaves [26]. As transcriptional activators, some Type-B ARRs, such as ARR14 and ARR18, have been shown to form homodimers [75,132]. For the Type-A ARRs, ARR5 was shown to homodimerize both in vitro and in vivo [133]. Regarding the two Type-C ARRs, ARR22 and ARR24, no interaction studies have been reported which could show if these two proteins are also able to dimerize.

Dimerization in bacteria is a widespread mechanism that alters DNA binding and target gene specificity, finalizing the signal output. Whether plant TCS proteins exert similar functions is still to be determined.

#### 4.2.2. Hot Spot Amino Acids Are Responsible for the Protein Binding Affinity

It cannot be denied that protein–protein interactions build a bridge and act as a junction for the phosphorelay cascade, but the protein binding efficiency might be more delicate to control the specificity of TCS signaling. Increasing evidence has shown that the interaction interfaces determined by some specific hot spot amino acids are responsible for the binding affinities, further affecting the strength of the protein–protein interaction.

To reveal hot spot amino acids in the interaction interfaces determining binding affinities between cytokinin receptors and HPts, virtual alanine scanning was performed and all 13 AHK-HPts modeled complexes were investigated. 3 hot spot positions, K1013, N898, and N901 (AHK3 numbering) were found in all the studied complexes, and two highly conserved hot spots, Q83 and S87, were also revealed in HPts [134]. CKI1 was confirmed to interact with AHP2, AHP3, and AHP5 both in vitro and in vivo, but ELISA results showed strong interactions with AHP2 and AHP3, and weaker interaction with AHP5 [77]. The main structural difference in the cytosolic parts is the absence of a receiver-like domain between CKI1 and cytokinin receptors. Arkhipov et al. compared the differences in the RD–HPt interaction interface, and the results showed that N901 (AHK3 numbering) is a hot spot replaced by a serine in CKI1 (S997, CKI1 numbering), leading to weaker interactions with some HPts [134]. AHK5 could interact with all the HPts except for AHP4, and the binding affinities of AHK5 with AHP1–3 were very similar, which correlates well with the interaction relationships of AHK2–4 with AHP1–3 [26]. Further investigations revealed that N901 (AHK3 numbering) was not replaced in AHK5 (N789, CKI2 numbering), unlike in CKI1, indicating that this amino acid position might serve as a hot spot amino acid for RD–HPt interactions [134]. Verma et al. investigated putative hot spot amino acids in the AHP1-ARR4 interaction. Using homology modeling, they suggested that 5 amino acids (D45, R51, Y96, C97, and P148) in ARR4 are responsible for protein interactions with AHP1. Two of these residues, D45 and Y96, were further experimentally confirmed as the interaction determinants [128]. Theoretically, hot spot amino acids exist in each protein-protein interaction interfaces, only a small fraction of these specific sites have already been identified. To further reveal the mechanisms of protein-protein binding affinity and TCS signaling specificity, more attempts should be required to identify these potential sites. For that purpose, manual hot spots-mimicking point mutation, structural biology and computational modelling tools may contribute to address these issues.

#### 4.2.3. Mg^2+^-Mediated Protein Binding Affinity

The activity of a wide range of proteins is mediated and regulated by ion metal binding [135]. Rec domain-containing proteins such as hybrid HKs or RRs require a magnesium ion in the active site to catalyze the phosphotransfer reaction and autocatalytic hydrolysis of phosphorylated aspartate. Pekarova et al. studied the effects of Mg^2+^ on the binding affinity of the CKI1-AHP complex. Results showed that adding Mg^2+^ increased the binding affinity of the CKI1_RD_–AHP2 complex, while adding Mg^2+^ led to a slight decrease in the affinity of the CKI1_RD_–AHP3 complex [77]. Analysis of AHK3_RD_-AHP2 dimer models in different phosphorylation states showed that Mg^2+^-bound TCS proteins markedly increased their mutual affinity in comparison to the apo-form. An increased binding affinity was observed when D941 of AHK3 was shifted to the phosphorylated state in the presence of Mg^2+^, while the binding affinity of AHK3_RD_-AHP2 dropped to a level even lower than that of the apo-forms [134]. The necessity of divalent metal ions is not restricted only to Mg^2+^, there are signs that some other metal ions, such as Mn^2+^, Ca^2+^, and Cd^2+^, might also have an important regulatory role in protein binding affinity. However, the molecular mechanisms regarding on the exact binding site and reaction kinetic are still largely to be determined.

### 4.3. Protein Modifications of Two Component System Elements

Protein modification is the last line of defense to finetune TCS signaling. To control signaling specificity, dephosphorylation and degradation of TCS protein modifications are efficient means to attenuate the signaling cascade or even switch them off, regardless of the presence of proteins and protein–protein interactions (Figure 9).

#### 4.3.1. Specificity of TCS Signaling via Protein Dephosphorylation

Protein dephosphorylation is a system for plants to regulate the specificity of TCS signaling. Some HKs in *Arabidopsis* have been shown to possess phosphatase activity. The cytokinin receptor AHK4 plays a dual role in cytokinin signaling, which acts as a kinase in the presence of cytokinin, and de-phosphorylates AHPs in the absence of cytokinin [130]. Similar to AHK4, truncated CKI1 protein with only the Rec domain showed phosphatase activity and reverse-phosphorylation activity toward AHP1 and AHP2 in vitro [136]. Dephosphorylation of AHPs, Type-A and Type-B ARRs have so far not been identified, but a Type-C ARR, ARR22, was proved to possess the ability to dephosphorylate AHP5 in a phosphorelay-dependent manner [114].

Another factor that should be considered as an important mechanism for signaling specificity is the phosphorylation lifetime of the aspartate residue. Studies in bacteria have shown that the phosphorylation lifetime varies in a range from seconds to hours and is directly influenced by various amino acid residues at defined positions within the Rec domain structure [137]. Several experiments have been carried out to elucidate the phosphorylation lifetime of ARRs in *Arabidopsis*. Upon phosphorylation, ARR4 remains in a phosphorylated state for 5 min, whereas ARR3 and ARR11 could remain in this state for up to 10 min [138,139]. When radioactively phosphorylated AHP5 was incubated with ARR22 in vitro, the radioactive phosphoryl group on AHP5 disappeared within only 1 min, and ARR22 was not phosphorylated [114], suggesting that the phosphorylation lifetime of ARR22 is much shorter. The presence of RRs with different phosphorylation lifetimes in the cell might contribute to canalizing phosphate flow into specific branches within the TCS signaling network.

#### 4.3.2. Specificity of TCS Signaling via Protein Degradation

Similar to bacteria, protein degradation of some TCS components mediated by the ubiquitin-proteasome system in plants represents a supplementary mechanism to regulate the specificity of TCS signaling. To date, only some Type-A and Type-B ARRs have been found to be targeted for proteolysis in *Arabidopsis*, but this process has not been reported for AHKs, AHPs, and Type-C ARRs. In case of Type-B ARRs, ARR2 was the first protein reported to be degraded by the 26S proteasome to modulate cytokinin signaling outputs. Kim et al. found that ARR2-hemagglutinin (HA) is rapidly degraded by cytokinin treatment, and degradation of ARR2 is mediated by the 26S proteasome pathway [140]. Except for ARR2, other Type-B ARRs, such as ARR1, ARR10, and ARR12, have been also reported as targets of the ubiquitin-proteasome system. Further analysis showed that these proteins could interact with KISSME DEADLY (KMD) proteins, suggesting that protein degradation mediated by the SCF^KMD^ complex negatively regulates the TCS signaling pathway by eliminating some Type-B ARR-positive regulators [141]. In addition to Type-B ARRs, two Type-A ARRs, ARR5 and ARR7 have been found to be targeted for proteolysis; however, since ARR7 could not interact with KMD proteins [141], a different recognition mechanism responsible for ARR7 degradation through the ubiquitin-proteasome system is still to be determined. The results also revealed that protein proteolysis mediated by ATP-independent serine protease DEG9 is another mechanism of protein degradation for a Type-A ARR, ARR4 [142]. Protein degradation of RRs creates an additional approach to fine-tune the specificity of TCS signaling. The ubiquitin-proteasome system and proteolysis system may represent the two main mechanisms for RRs protein degradation, but not confined to them. As for other undescribed RRs involved in the TCS pathway, whether they undergo the same protein degradation process and the hidden degradation mechanisms remain largely unknown.

## 5. Conclusions and Perspectives

In summary, this review focuses on recent advances in the TCS-mediated MSP signaling pathway in the model plant *Arabidopsis*. The protein–protein interaction information was retrieved and an interaction map of AHK-AHP and AHP-ARR has been depicted. Subsequently, the possible molecular mechanism of MSP signaling specificity at 3 different levels is discussed. At the mRNA level, gene expression specificity of TCS represents the primary level to fine-tune the signaling specificity in the plant MSP network. Then, a homodimeric or a heterodimeric complex formed among the TCS protein elements at the protein level ensures the correct phosphate transfer and signal output. Factors influencing protein binding affinities, such as hot spots amino acids and metallic ion Mg^2+^, play essential roles in determining the efficiency and specificity of signal transduction. Dephosphorylation and degradation of TCS protein elements at the protein modification level could attenuate the signaling cascade or even switch them off, further contributing to the specificity of MSP signaling. Overall, these 3 levels of molecular mechanisms fine-tune the signaling specificity of the *Arabidopsis* MSP network in a collaborative and indispensable manner.

In this review, we established an interaction map of TCS proteins based on current interaction relationship studies. However, massive information is still lacking in the map. The AHKs–AHPs interactions were better studied, whereas interactions of AHPs–ARRs are still poorly understood. Not all ARR proteins were separately surveyed to identify their AHP interaction partners, and interaction relationships for several ARR members were not studied at all; thus, more evidence is necessary to fulfill the interaction map. Besides, as the final step of phosphorylation, Type-B ARRs are downstream partners which act as transcription factors to regulate common targets in a redundant manner or regulate some unique targets. The knowledge on the downstream targets so far is still limited. In addition to the well-established model that cytokinin and ethylene act their roles via the TCS, other phytohormones, such as auxin, abscisic acid, and gibberellin may also crosslink with TCS, further increasing complexity of the TCS interaction map.

The three-dimensional crystal structures of TCS elements could provide important information for studying protein binding characteristics, especially in the process of illuminating the MSP signaling specificity mechanism. Currently, a large number of three-dimensional crystal structures of prokaryotic and eukaryotic TCS elements, mainly determined by X-ray and nuclear magnetic resonance (NMR), have been reported. However, information on the crystal structure of TCS components in higher plants is still limited. To date, only structures have been solved for some HKs (ETR1, CKI1, AHK4, and AHK5) and two HPts, AHP1, and AHP2, in addition to some HPts in maize, rice, and alfalfa [77,78,143,144,145,146,147]. Further high-resolution crystal structure characterization of TCS elements could provide more clues to improve our understanding of the signaling specificity of the plant MSP network.

TCS elements were originally identified in *E. coli*, and the one-step TCS signaling pathway has been extensively studied and well understood in prokaryotes. Plants have evolved a more complicated MSP pathway, but still share many similarities with the bacterial TCS signaling machinery, basically, the nature of the phosphorelay is not essentially different. To gain more new insights into the evolved signaling mechanism of TCS both in prokaryotes and eukaryotes, Mira-Rodado described and compared the different regulatory processes that occur in the canonical bacteria TCS and MSP pathway in *Arabidopsis*, and summarized that protein degradation, protein de-phosphorylation and protein dimerization are 3 main regulatory mechanisms for both bacteria and plant to modulate TCS signaling specificity [11]. Since TCS signaling pathway was better understood in bacteria, thus, a comparative study with bacterial TCS might facilitate progress in the research efforts on plant TCS.

## Figures and Tables

**Figure 1 ijms-21-04898-f001:**
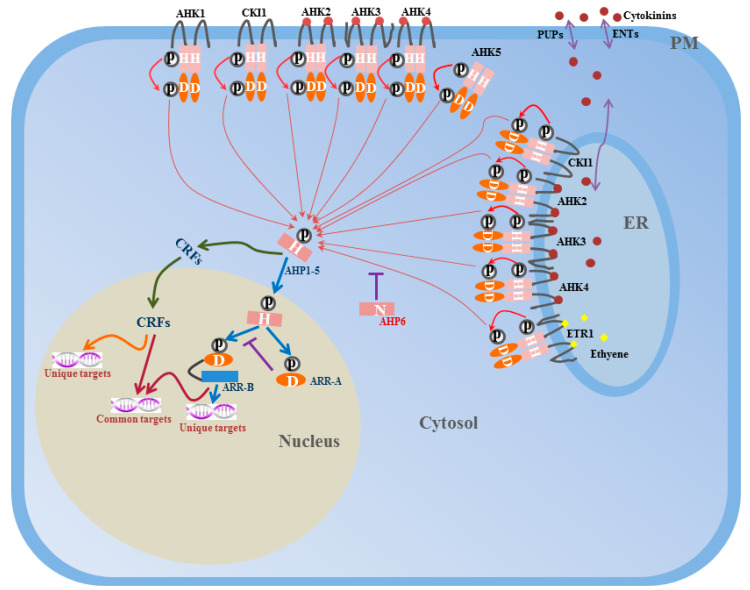
Model for two-component system signal transduction pathway in *Arabidopsis*. Signal perception induces autophosphorylation of the conserved histidine residue (H) in the histidine kinase (HK) domain of AHKs localized in the PM or ER. Then the phosphoryl group (P) is subsequently transferred to a conserved aspartate residue (D) of the C-terminal receiver domain. AHP proteins accept the signal from AHKs, and transfer the phosphoryl group to the ARR-A or ARR-B. AHP6 lacks the conserved histidine residue and negatively regulate phosphate transfer process. ARR-B are transcription factors which could positively regulate downstream targets while ARR-A negatively regulate ARR-B expressions. As a parallel branch, AHP proteins may also transfer the phosphoryl group to another transcription factors, cytokinin response factors (CRFs). Upon phosphorylation, CRFs shuttle from cytosol to the nucleus and regulate downstream targets. CRFs and Type-B ARRs share some common targets while also having their own specific targets. Cytokinin and ethylene are indicated by red dot and yellow diamond, respectively. PUPs (purine permeases) and ENTs (equilibrative nucleotide transporters) are cytokinin transporters. Figure indicates a whole cell. ER, endoplasmic reticulum; PM, plasma membrane; H, histidine residue; D, aspartate residue; P, phosphoryl group; ARR-A, Type A ARRs; ARR-B, Type B ARRs.

**Figure 2 ijms-21-04898-f002:**
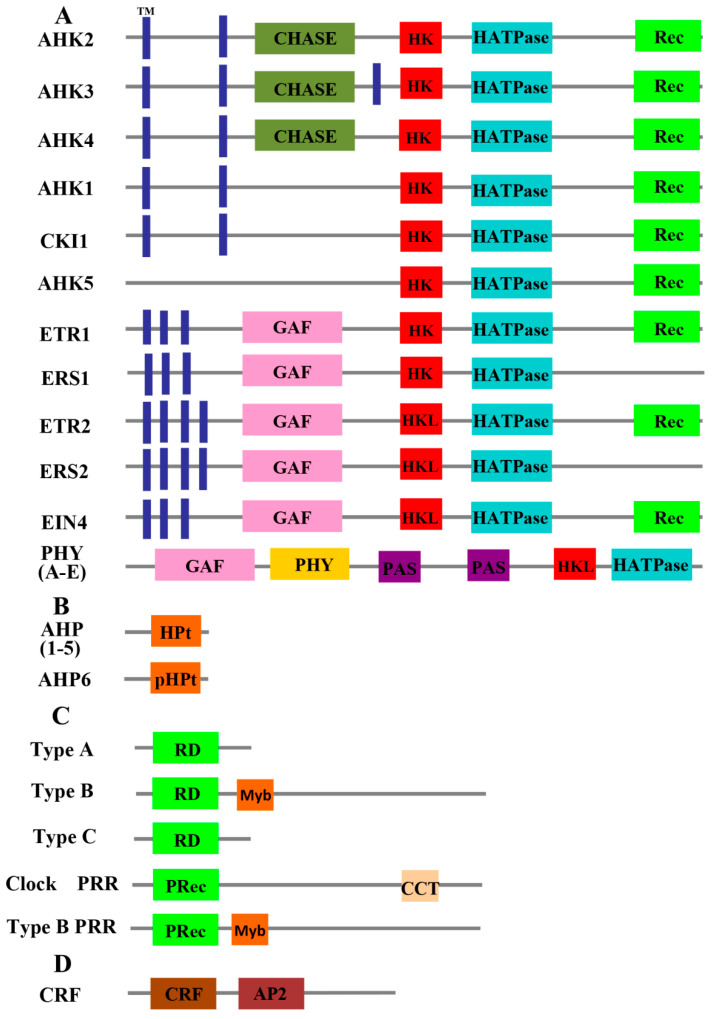
Representative domain structures of *Arabidopsis* two-component signaling elements. (**A**) Domain structures of HKs. (**B**) Domain structures of Hpts. (**C**) Domain structures of RRs. (**D**) Domain structures of CRFs. TM, transmembrane; CHASE, cyclases/histidine kinases associated sensing extracellular; HK, histidine kinase; HATPase, histidine adenosine triphosphatase; Rec, receiver; GAF, cGMP-regulated phosphodiesterases and adenylate cyclases of *Anabaena* and the bacterial transcription factor FhlA; HKL, histidine kinase like; PHY, phytochrome; PAS, Per/Arndt/Sim; HPt, histidine-containing phosphotransfer; pHPt, pseudo histidine-containing phosphotransfer; RD, receiver domain; PRec, pseudo receiver; CCT, Co, Col and Toc1; CRF, cytokinin response factor; AP2, APETALA 2.

**Figure 3 ijms-21-04898-f003:**
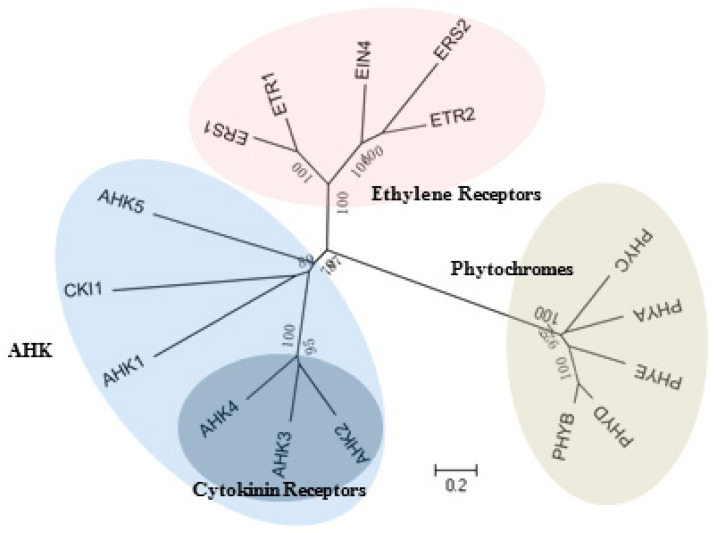
Phylogenetic relationship of HKs in *Arabidopsis*. The tree is branched into 3 groups: AHK family, ethylene receptor family and phytochrome family, 3 cytokinin receptors located in a subdivided branch of AHK family.

**Figure 4 ijms-21-04898-f004:**
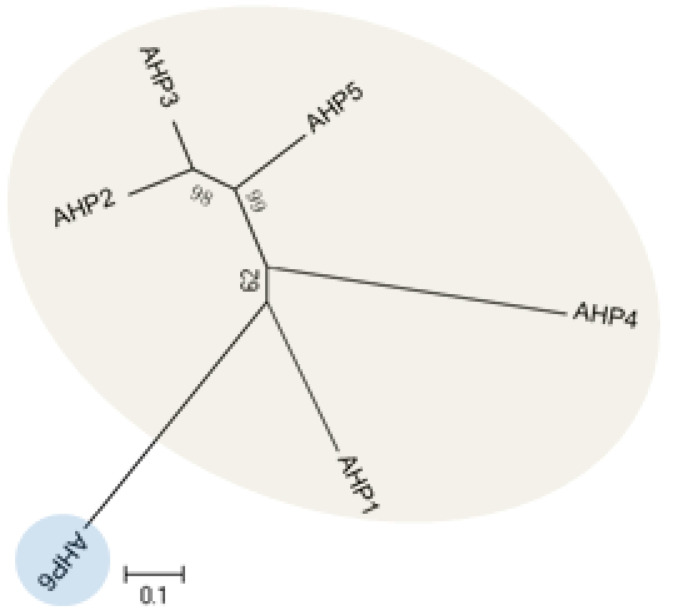
Phylogenetic relationship of histidine-containing phosphotransfer proteins (HPts) in *Arabidopsis*.

**Figure 5 ijms-21-04898-f005:**
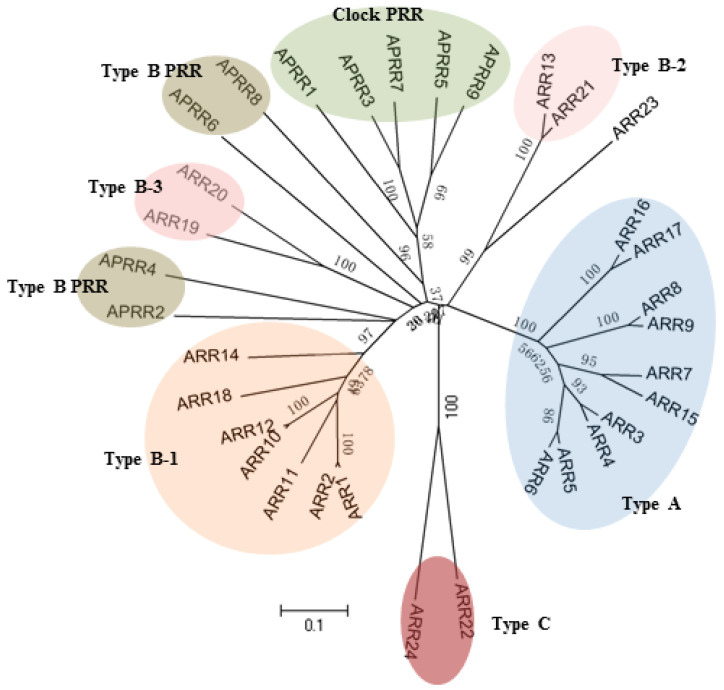
Phylogenetic relationship of response regulators (RRs) in *Arabidopsis*.

**Figure 6 ijms-21-04898-f006:**
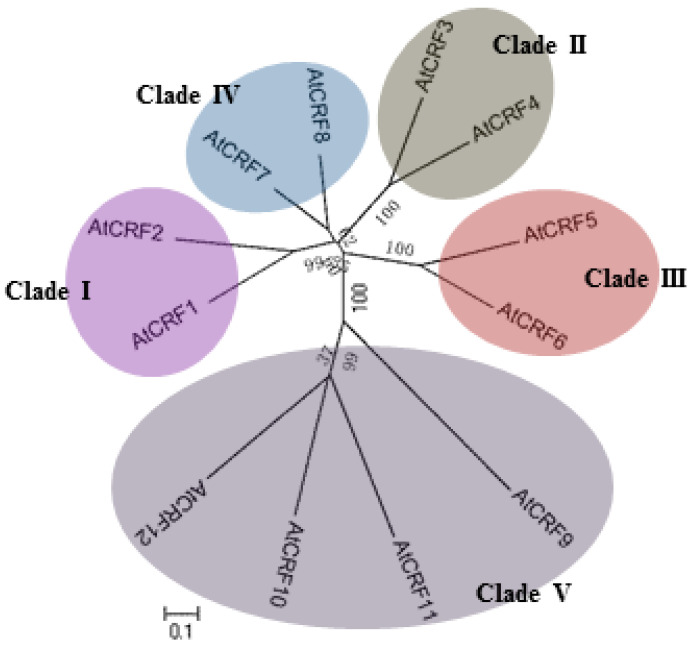
Phylogenetic relationship of CRFs in *Arabidopsis*.

**Figure 7 ijms-21-04898-f007:**
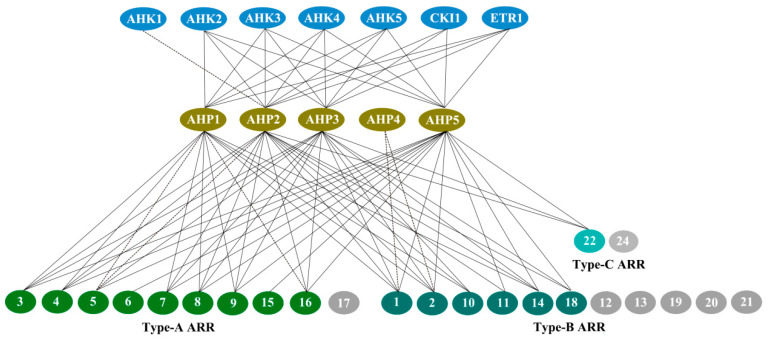
Protein–protein interaction network of two-component systems (TCS) signaling pathway in *Arabidopsis*. Interactions confirmed both in in vitro and in vivo are indicated by a line, while interactions found only in the Y2H analysis are indicated by dashed lines. Shaded oval shapes represent proteins not studied.

**Figure 8 ijms-21-04898-f008:**
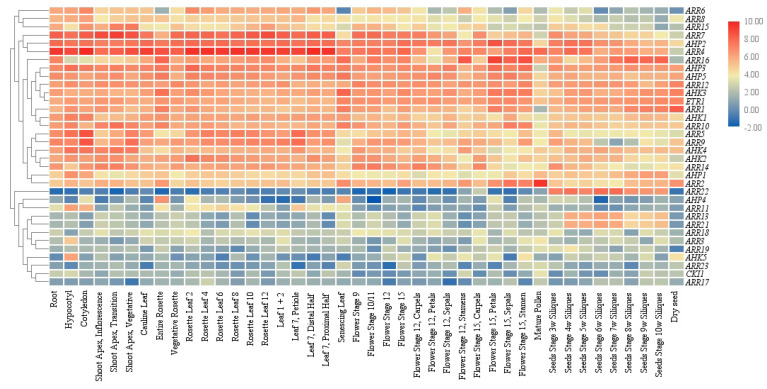
Hierarchical clustering and heat map representation for the TCS gene expression profiles in *Arabidopsis*. Gene expression data were obtained from the AtGenExpress Consortium (*Arabidopsis* eFP Browser).

**Figure 9 ijms-21-04898-f009:**
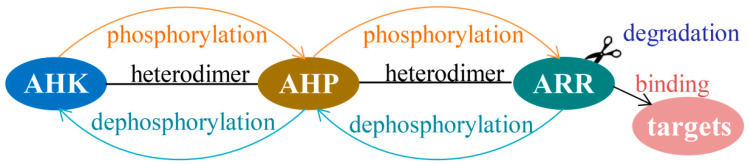
Model of TCS signaling specificity in *Arabidopsis*.

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
