# Peer review of "The Interaction Network and Signaling Specificity of Two-Component System in Arabidopsis"

_ijms, 2020, doi:10.3390/ijms21144898_

Round 1
Reviewer 1 Report
The Review entitled "The interaction network and signaling specificity of two-component system in Arabidopsis" had as objective was to present a complete overview of the updated knowledge on two-component system in Arabidopsis, emphasizing signal transduction.
As the authors describe, the literature reports many information’s on the role of this proteins in signaling roles. Hence the importance of the review which reports in a very well manner some molecular aspects of the role of these mechanisms. The figures and schemes show a good review and are very well descriptive; the language is clear and the text is well structured. The purpose of the review is to lay the foundations for future understandings of the resource for research focused on the development of signaling mechanisms in Arabidopsis. The review, however, must be improved in terms of writing since some grammar and syntax errors are present in the manuscript. In addition, it would be interesting to present a topic better reporting the mechanisms of evolution of the two-component systems in general, and then going to plants, because in our opinion the central topic is the two-component system. In we opinion the Authors only partly succeeded in their intent. In some parts of the manuscript, the same looks like a mere list of references, it is not a critical review. In particular, concluding remarks are based only on general considerations on the system. They should address the subject and critically review the information from the literature.
Author Response
Dear Reviewers:
We would like to express our earnest gratitude for your affirmation and constructive proposals to our manuscript (ijms-848253) entitled “The interaction network and signaling specificity of two-component system in Arabidopsis”. All the authors read and discussed your comments and suggestions, we tried to revise and improve the manuscript according to your constructive suggestions. All the revisions were indicated by red font to be easily visible to you.
The main corrections and the responses to the reviewer’s comments are as follows:
(1) The review, however, must be improved in terms of writing since some grammar and syntax errors are present in the manuscript.
Response: We applied for the Wiley editing services to polish this manuscript. And in this revised manuscript, we further checked the language with some possible grammar and syntax errors. All the revisions were indicated by red font to be easily visible to you.
(2) In addition, it would be interesting to present a topic better reporting the mechanisms of evolution of the two-component systems in general, and then going to plants, because in our opinion the central topic is the two-component system. In we opinion the Authors only partly succeeded in their intent.
Response: It is truly interesting to discuss the mechanisms of TCS evolution, in fact we planned to include this part when we first draft the manuscript, but gave it up for two considerations. 1) In our lab, we have analyzed the evolution of HK proteins in various genomes including both the prokaryotes and eukaryotes, and screened some hot spots amino acids, and now we are continuing with this project to reveal the possible roles of these sites with point mutation or domain swap techniques, and we prefer to discuss the evolution in this upcoming manuscript. So we only give a brief description of one-step TCS in prokaryotes and multi-step TCS in plants in the introduction part. 2) In some previous review articles, the authors discussed the evolution of TCS from bacteria to higher plants, please refer to reviews by Mira-Rodado, 2019; Schaller et al., 2008; Jacob-Dubuisson et al., 2018.
Of course, we would respect your considerations if you insist it is better to include some paragraphs discussing the TCS evolution.
(3) In some parts of the manuscript, the same looks like a mere list of references, it is not a critical review. In particular, concluding remarks are based only on general considerations on the system. They should address the subject and critically review the information from the literature.
Response: It is true that in the manuscript we emphasized more on the recent advances on Arabidopsis TCS, and discussions in some part of the paragraph are insufficient. In the revised manuscript, we supplement discussions mainly in part 4 (e.g. 4.2.2 and 4.2.3) and part 5 which indicated by red font. We hope the supplements could further improve the manuscript.
Reviewer 2 Report
This manuscript represents a very comprehensive review on the plant (Arabidopsis thaliana) version of two-component signalling systems. The main types/families of these regulatory mechanisms are discussed citing relevant and current literature. The topic is huge, since these phosphorelay-based switches are involved in the modulation of numerous hormonal and light signaling pathways. Description of these signalling pathways, at least the one dealing with my special field, is correct and free of overstatements. This report is not a simple collection of available data, but results are discussed properly and moreover, original evaluation and interpretation of published data is provided.
I have only a few comments/suggestions, the consideration of which could improve the text of the paper.
- For all figure legends: all abbreviations should be explained. For Figure 1: a more detailed description of the processes shown in the figure should be provided.
- Page 4, paragraph of the AHK family: explain ‘Rec domain’. In general, all abbreviations should be explained at the first mention.
- Page 8. The text says: “Type-A ARRs function as negative regulators of cytokinin signalling pathways”. Later the authors say that the type-B ARR6 positively regulates the expression of some type A factors. So, the suggestion that “Type-B ARRs function as positive regulators of cytokinin signalling “looks contradictory.
- Page 8: “… transcripts of Type -B ARR genes were stable…) Do you mean stable = transcription rate, or stability/turnover of the mRNA? Please clarify.
- Page 9. “… Type-B ARRs may be direct targets of CFRs. Previous data says the opposite: CFRs can be direct targets of type B ARRs. Please correct it.
Author Response
Dear Reviewers:
We would like to express our earnest gratitude for your affirmation and constructive proposals to our manuscript (ijms-848253) entitled “The interaction network and signaling specificity of two-component system in Arabidopsis”. All the authors read and discussed your comments and suggestions, we tried to revise and improve the manuscript according to your constructive suggestions. All the revisions were indicated by red font to be easily visible to you.
The main corrections and the responses to the reviewer’s comments are as follows:
(1) For all figure legends: all abbreviations should be explained. For Figure 1: a more detailed description of the processes shown in the figure should be provided.
Response: We defined the abbreviations in Figure 1 and Figure 2. For the Figure 3-9, if the abbreviations were defined previously, abbreviations were used. Besides, a description of MSP signaling pathway of Arabidopsis was provided in Figure 1.
(2) Page 4, paragraph of the AHK family: explain ‘Rec domain’. In general, all abbreviations should be explained at the first mention.
Response: We checked the abbreviations in the whole manuscript, and abbreviations were defined when first appeared.
(3) Page 8. The text says: “Type-A ARRs function as negative regulators of cytokinin signalling pathways”. Later the authors say that the type-B ARR6 positively regulates the expression of some type A factors. So, the suggestion that “Type-B ARRs function as positive regulators of cytokinin signalling “looks contradictory.
Response: Truly it looks contradictory, in the revised manuscript, “Type-B ARRs function as positive regulators of cytokinin signalling” was replaced with “Type-B ARRs function as positive regulators in mediating some Type-A ARR gene expression”.
(4) Page 8: “… transcripts of Type -B ARR genes were stable…) Do you mean stable = transcription rate, or stability/turnover of the mRNA? Please clarify.
Response: In the revised manuscript, “transcripts” was replaced with “mRNA transcripts”, to make it clear.
(5) Page 9. “… Type-B ARRs may be direct targets of CFRs. Previous data says the opposite: CFRs can be direct targets of type B ARRs. Please correct it.
Response: Thanks for point out this error. In the revised manuscript, we changed the sentence to “CRFs may be the direct targets of Type-B ARRs”.
Round 2
Reviewer 1 Report
Accept in present form